# Skeletal metalation of lactams through a carbonyl-to-nickel-exchange logic

Hongyu Zhong [1], Dominic T. Egger [1], Valentina C. M. Gasser[1,2], Patrick Finkelstein[1,2], Loris Keim[1], Merlin Z. Seidel [1], Nils Trapp[1] & Bill Morandi [1] ✉

Classical metalation reactions such as the metal-halogen exchange have had a transformative impact on organic synthesis owing to their broad applicability in building carbon-carbon bonds from carbon-halogen bonds. Extending the metal-halogen exchange logic to a metal-carbon exchange would enable the direct modification of carbon frameworks with new implications in retrosynthetic analysis. However, such a transformation requires the selective cleavage of highly inert chemical bonds and formation of stable intermediates amenable to further synthetic elaborations, hence its development has remained considerably challenging. Here we introduce a skeletal metalation strategy that allows lactams, a prevalent motif in bioactive molecules, to be readily converted into well-defined, synthetically useful organonickel reagents. The reaction features a selective activation of unstrained amide C−N bonds mediated by an easily prepared Ni(0) reagent, followed by CO deinsertion and dissociation under mild room temperature conditions in a formal carbonyl-to-nickel-exchange process. The underlying principles of this unique reactivity are rationalized by organometallic and computational studies. The skeletal metalation is further applied to a direct CO excision reaction and a carbon isotope exchange reaction of lactams, underscoring the broad potential of metal-carbon exchange logic in organic synthesis.

Organometallic reagents have been a constant powerhouse in organic synthesis. For the metal-halogen exchange reaction, the facile preparations of Grignard[1] and organolithium reagents[2] directly from the widely available organic halides in a formal umpolung process have enabled their coupling to a broad range of electrophiles. In contrast to halogens which are typically monovalent atoms, carbons are tetravalent atoms that ubiquitously constitute the backbones of organic molecules. An analogous metal-carbon exchange reaction would be poised to greatly impact organic synthesis by introducing new disconnection approaches[3]. Through the instalment of reactivity hotspots onto carbon frameworks, such a "skeletal metalation" reaction could facilitate the direct remodelling of ring structures (Fig. 1a). The identification of a general skeletal metalation reaction could find immediate applications in medicinal chemistry for lead structure diversifications[4], whereby a custom-made organometallic intermediate synthesised from the lead compound could be rapidly transformed into multiple skeletally modified analogues that would normally require laborious de novo synthesis. However, a metal-carbon exchange is considerably challenging because (1) not only one, but two inert bonds need to be selectively cleaved; (2) the structural integrity of the rest of the molecule needs to be maintained. Although the release of ring strain and/or high reaction temperature are established strategies that address the reactivity challenge[5–7], they are typically tied to ring expansion transformations, while related examples on alternative single-atom editing such as ring contraction have remained particularly rare[8–11]. Expanding such transformations

[1]Laboratorium für Organische Chemie, ETH Zürich, 8093 Zürich, Switzerland. [2]These authors contributed equally: Valentina C. M. Gasser, Patrick Finkelstein. ✉e-mail: morandib@ethz.ch

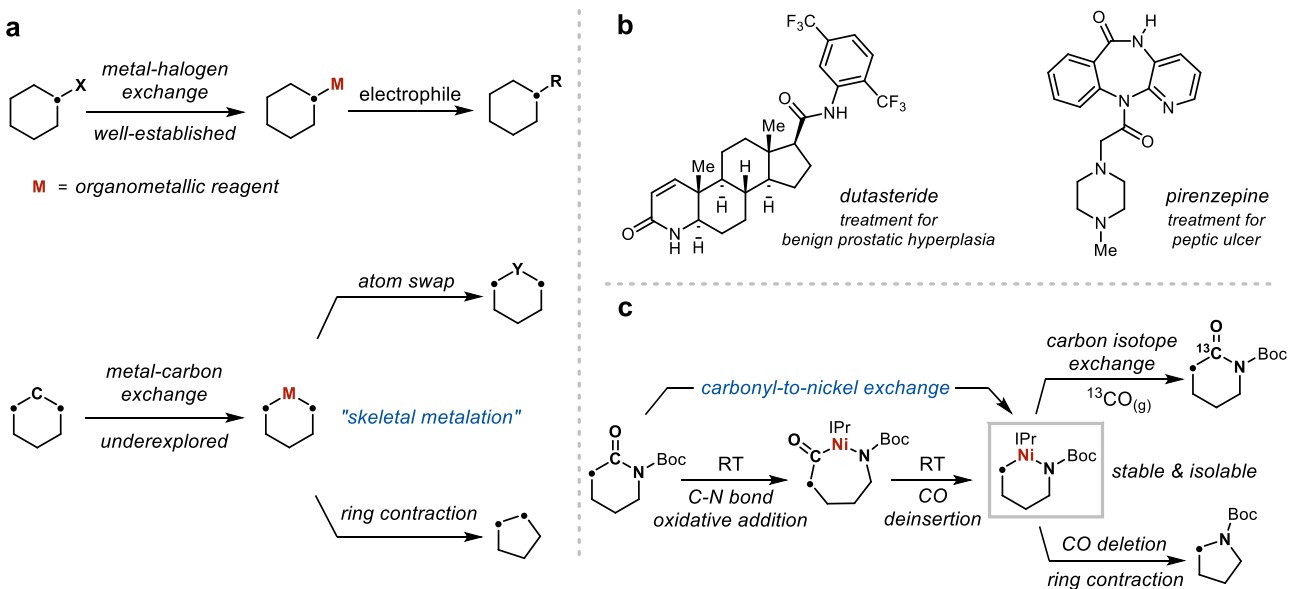

**Fig. 1 | Skeletal metalation of lactams. a** Metal-carbon exchange provides new opportunities for molecular editing. **b** Examples of pharmaceutical compounds with a lactam core. **c** This work: a carbonyl-to-nickel-exchange strategy for skeletal editing of lactams.

to more versatile, C(sp³)-rich targets will reveal the full potential of metal-mediated single-atom editing in drug-like molecules.

The lactam ring is a ubiquitous core structure in pharmaceutical compounds, antibiotics and natural products (Fig. 1b)[12,13]. Although reactions such as carbonyl reduction or reductive functionalisation of the amide bond of lactams have been well established[14,15], the direct editing of the carbon framework, for example, the excision of the carbonyl moiety as a CO molecule giving a one-carbon-contracted cyclic amine product, has remained elusive. Such a reaction would also enable a lactam (ring size = $n$) to serve as a "masked" cyclic amine (ring size = $n−1$) that could be carried through multiple synthetic steps before revealing itself through a formal CO deletion reaction. Thus, developing a skeletal metalation platform for lactams, in which the carbonyl moiety could be directly exchanged by a metal, could facilitate such a transformation and even unlock more versatile atom swap or ring expansion processes that are particularly attractive in the structural remodelling of organic molecules[16–25].

Here we introduce a carbonyl-to-metal-exchange logic for the skeletal metalation of 4- to 8-membered lactams. The reaction employs an easily prepared Ni(0) precursor and occurs readily at room temperature, affording a versatile new class of well-defined organonickel reagents. Experimental and computational studies identified the effects from nitrogen substituents and lactam ring size on the C(O)−N bond activation, CO deinsertion and dissociation processes, which constitute the key steps for the carbonyl-to-metal exchange. These insights gained from organometallic studies are applied to the skeletal editing of lactam cores in medicinally relevant molecules, including a net CO deletion reaction and a carbon isotope exchange reaction of lactam carbonyls using ¹³CO(g) (Fig. 1c), clearly highlighting the synthetic potential of the carbon-to-metal-exchange logic.

## Results and discussion

Inspired by the pioneering work on transition metal-catalysed activations of linear amides, esters and other carbonyl compounds[26–47], we commenced our study by investigating the organometallic chemistry of oxidative addition of endocyclic amide C−N bonds to an electron-rich, low-valent metal centre. The N-heterocyclic carbene (NHC)-ligated Ni(0) complexes have shown high activity towards inert bond activations[26,27,29,38,41,43–45,48,49]. In particular, the well-defined (NHC)Ni(0)(η⁶-arene) complexes[50,51] with substitutionally labile arene ligands are

suitable sources of a highly coordinatively unsaturated "(NHC)Ni(0)" active species that might favourably bind and activate a lactam without potential inhibitory equilibriums from coordination of other ancillary ligands. In our preliminary computational survey of the relative Gibbs free energy of (IPr)Ni(0)(L)ₙ complexes (IPr = 1,3-Bis(2,6-di-i-propyl-phenyl)imidazol-2-ylidene, L = ancillary ligand), (IPr)Ni(0)(η⁶-PhMe)[50] showed the highest relative reactivity and drove our initial experiments (Supplementary Fig. 1). Several challenges can be expected for the C−N bond oxidative addition of lactams. First, a selective activation of the desired endocyclic C−N bond affording a metallacycle needs to occur, while an exocyclic activation giving a potentially stabilised κ²-amidate complex[52] can be competing (Fig. 2a). Upon formation of the metallacycle, the complete deinsertion and dissociation of CO from the metal could prove difficult especially for a late transition metal that forms strong π back-bonding to CO ligands[53]. For metallacycles derived from aliphatic lactams, ring opening by β-hydride elimination may also decrease their stability with potential influences from ring size and temperature[54]. To preclude undesired oxidative addition of the N−H bond to Ni(0)[55], the commonly used Boc (tert-butyloxycarbonyl) protecting group[56] is employed, which has also been shown to weaken the amide bonds by electron-withdrawing and geometric twisting effects and N-Boc amides have been widely used in transition metal catalysis[27–29,31,38,43,44,46,57]. Our study also sheds light on the role of the Boc group in mediating the elementary steps of C−N bond oxidative addition and CO deinsertion (vide infra).

When γ-lactam **1** was mixed with one equivalent of (IPr)Ni(η⁶-PhMe) in toluene-$d_8$ under argon atmosphere, a new diamagnetic nickel species **Ni-1** was formed in 24% conversion after 5 h at room temperature (RT) monitored by ¹H NMR spectroscopy (Fig. 2b). After reaching 69% conversion at 48 h, **Ni-1** slowly underwent CO deinsertion affording a nickel alkyl species **Ni-1′** (25% conv. after 10 days at RT) indicated by ¹H NMR spectroscopy. Isolation of single-component **Ni-1** on preparative scale and characterisation by ¹H, ¹³C{¹H} NMR, IR spectroscopy and X-ray crystallography established its identity as the desired nickel(II) product from the endocyclic activation of lactam **1**, providing a rare structural evidence of a nickel(II) acyl complex directly from oxidative addition of an amide C−N bond[43,58]. The X-ray structure of **Ni-1** shows a pseudo square planar geometry at the nickel centre featuring a κ²-N,O coordination mode from the N−Boc moiety, identifying a role of the Boc group in stabilising the nickel intermediate

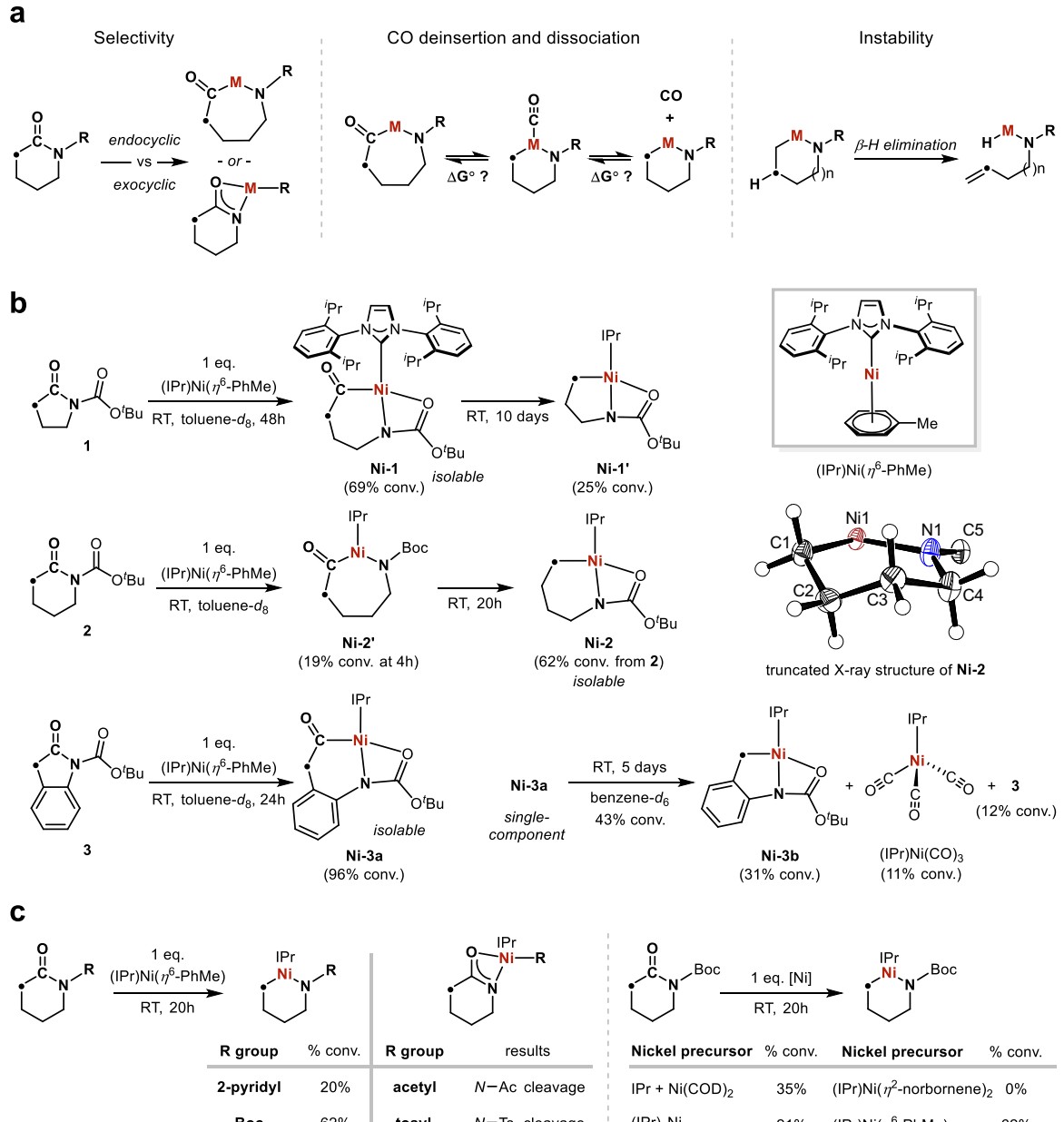

**Fig. 2 | Design elements and initial observations. a** Challenges in selectivity, unfavourable CO deinsertion and dissociation, and ring-opening β-hydrogen elimination for developing the skeletal metalation of lactams. **b** Initial discoveries of C–N bond oxidative addition and CO deinsertion trends with γ- and δ-lactams. Truncated X-ray structure depicted with 50% thermal ellipsoids. **c** Evaluation of alternative nitrogen substituents and nickel precursors. Reactions performed on 0.1 mmol scale.

by making a four-coordinate complex. Because the CO deinsertion of **Ni-1** was considerably slow, the nickel alkyl **Ni-1′** proved inseparable from **Ni-1**.

A much faster CO deinsertion was observed when the lactam ring contains one extra carbon atom. When δ-lactam **2** was treated with one equivalent of (IPr)Ni(η⁶-PhMe) at RT, the CO-deinserted product **Ni-2** was successfully isolated, while the nickel acyl intermediate **Ni-2′** was too unstable to intercept (Fig. 2b). Remarkably, the 6-membered **Ni-2** with four contiguous methylene groups was stable to β-H elimination at RT, indicating the potential for skeletal metalation of saturated δ-lactam cores that are common motifs in pharmaceutical compounds[13].

Insights into the thermodynamic driving force for the CO deinsertion and dissociation are obtained from the studies of the acyl nickellacycle **Ni-3a** synthesised from lactam **3** (Fig. 2b). Upon standing in a benzene-$d_6$ solution at RT for 5 days, 43% of **Ni-3a** converted into a

mixture of the deinserted nickel product **Ni-3b** (31%), (IPr)Ni(CO)$_3$[59] (11%) and lactam **3** (12%). This unusual observation suggests that the oxidative addition of the amide C(O)–N bond is reversible at room temperature indicated by the formation of lactam **3** from **Ni-3a**. Moreover, generation of the thermodynamically low-lying (IPr)Ni(CO)$_3$ from (IPr)Ni(0)(L$_n$) likely drives the CO deinsertion and dissociation from **Ni-3a** by capturing the liberated CO.

The successful skeletal metalation of lactams **1**, **2** and **3** with (IPr)Ni(η⁶-PhMe) prompted the investigation into different nitrogen substituents and nickel precursors. By employing 2-pyridyl as the directing group, the corresponding nickellacycle was also obtained albeit in lower yield (Fig. 2c). The N–acetyl and N–tosyl groups were found to be incompatible because the undesired exocyclic activation was favoured leading to nickel lactamate products (Supplementary Fig. 7). Alternative nickel precursors including (IPr)Ni(η²-norbornene)$_2$[60], (IPr)$_2$Ni[61]

**Fig. 3 | Metalation of 4- to 8-membered lactams.** Formation of organonickel reagents with isolated yields. Truncated X-ray structures depicted with 50% thermal ellipsoids.

or an equimolar mixture of IPr and Ni(COD)$_2$ showed lower activity compared to (IPr)Ni($\eta^6$-PhMe) corroborating the results from our computational investigations. Thus, the use of the Boc protecting group and the (IPr)Ni($\eta^6$-PhMe) reagent was maintained for the rest of our studies.

The generality of the skeletal metalation reaction was demonstrated with 4- to 8-membered lactams with diverse substitution (Fig. 3), affording nickellacycles **Ni-4** to **Ni-15** that have been fully characterised by 1 and 2-dimensional NMR and IR spectroscopy as well as X-ray crystallography. Metalation of the β-lactam **4** furnished the acyl nickellacycle **Ni-4** with no CO deinsertion. For γ-lactams **5**, **6** and **7**,

the corresponding nickellacycles **Ni-5, Ni-6** and **Ni-7a** were isolated. While **Ni-5** and **Ni-7a** were thermally unstable, heating at 60 °C was required for the metalation of **6** which directly afforded the deinserted product **Ni-6**. Slow CO deinsertion of **Ni-7a** at RT led to **Ni-7b** similar to the case of **Ni-3**. For δ-lactams **8**, **9**, **10**, **11** and **12**, the CO-deinserted nickellacycles were invariably isolated, suggesting that more facile carbonyl-to-nickel exchange occurs with δ-lactams compared to γ-lactams. Various substitution patterns were compatible for the metalation reaction, including an internal olefin (**Ni-8**), a fused benzene (**Ni-9, Ni-10**) and a β-oxygen (**Ni-11, Ni-12**). The near-quantitative isolation of **Ni-12** from equimolar **12** and (IPr)Ni($\eta^6$-PhMe) compared to

the yield of 71% from lactam **9** also suggests the substitution of a methylene group for an oxygen atom at the $\beta$-position renders the carbonyl-to-nickel exchange more facile. For 7- and 8-membered lactams **13, 14** and **15**, the desired nickellacycles **Ni-13, Ni-14** and **Ni-15** were also isolated in high yields. The 7-membered **Ni-13** was stable to $\beta$-hydrogen elimination after 40 h at RT, and **Ni-14** was stable to air in the solid state after 10 days. The activation of **14**, which possesses the core structure of the peptic ulcer medication pirenzepine[62], was also highly selective for the amide C–N bond in the presence of two other C(sp$^2$)–N bonds. Likewise, the 8-membered **Ni-15** was isolated in 84% yield. Further studies revealed that alkyl nickellacycles derived from 7-, 8- and 9-membered parent aliphatic lactams **16, 17** and **18** underwent increasingly faster $\beta$-hydride eliminations preventing their isolation (Supplementary Fig. 19), highlighting that the structural rigidity imposed by the fused aromatic rings in **13, 14** and **15** is key to their stability. These findings demonstrated a straightforward carbonyl-to-nickel exchange reaction of 6- to 8-membered lactams through the activation of unstrained amide bonds.

Density functional theory (DFT) studies were carried out to probe the energy landscape of the proposed elementary steps involved in the formation of the isolated nickellacycles (Fig. 4a)[63]. Representative lactams **1** and **2** were selected due to the contrasting isolation of the corresponding acyl- and alkyl nickellacycles **Ni-1** and **Ni-2**. The computed Gibbs free energy barriers for the chemoselective endocyclic C–N bond oxidative addition are 13.5 and 12.5 kcal/mol for **1** and **2** from the corresponding nickel(0)-lactam adducts **IM1** and **IM1'**, affording the energetically favoured acyl nickellacycles **IM2** and **IM2'** with relative energies of −12.5 and −15.9 kcal/mol. As the release of toluene from (IPr)Ni($\eta^6$-PhMe) is expected to be disfavoured in large excess of toluene as the solvent, a correctional term of roughly 3.5 kcal/mol under the concentration for the NMR-scale reactions was calculated, which would raise the barriers for C–N bond oxidative addition to 16.0 and 15.0 kcal/mol, respectively (Supplementary Figs. 2 and 3). The CO deinsertion barriers from **IM2** and **IM2'** were found to be 23.9 and 19.8 kcal/mol, respectively, affording intermediates **IM3** and **IM3'** with decoordination of the Boc carbonyl. Dissociation of the CO ligand from nickel and re-coordination of the Boc carbonyl afford the alkyl nickellacycles **P1** and **P1'** with relative energies of 7.1 and −2.8 kcal/mol. A limitation of our current model is the treatment of free CO in the reaction mixture, as its exact distribution between the liquid and gas-phase for our system is hard to approximate. The reversible coordination of the hemilabile Boc arm may be critical to promote CO deinsertion and dissociation dynamics in the square-planar Ni(II) complex[64]. Trapping of free CO by an unreacted (IPr)Ni($\eta^6$-PhMe) and formation of (IPr)Ni(CO)$_3$ would lower the energies of **P1** and **P1'** to −11.1 and −21.1 kcal/mol respectively. The DFT studies suggest a reversible carbonyl-to-nickel exchange scenario and formation of (IPr)Ni(CO)$_3$ or other (IPr)Ni-carbonyl species as a potential thermodynamic driving force that promotes the CO deinsertion and dissociation from the low-lying acyl nickellacycles **IM2** and **IM2'**. The contrasting isolation of the nickel alkyl **Ni-2** compared to the nickel acyl **Ni-1** may originate from the lower barrier (4.1 kcal/mol) for CO deinsertion and the higher stability (9.9 kcal/mol) of the CO-deinserted product **P1'**.

The microscopic reverse of the skeletal metalation reaction, namely the CO insertion and C(O)–N bond reductive elimination, readily occurred by adding exogenous CO$_{(g)}$ (Fig. 4b). Quantitative formation of starting lactams **6, 10** and **13** from the corresponding nickellacycles was established following addition of one atmosphere of CO$_{(g)}$, along with clean formation of (IPr)Ni(CO)$_3$ as the sole nickel species. This finding unveiled a protocol to exchange the lactam carbonyl with exogenous CO$_{(g)}$ with potential applications in carbon isotope labelling[65–67], highlighting a useful synthetic application of the organonickel reagents. Moreover, the reductive elimination of C(sp$^3$)–N or C(sp$^2$)–N bonds from the deinserted nickellacycles

was also investigated, as it would provide a net CO deletion process for single-atom editing. Oxidatively induced C(sp$^3$)–N reductive elimination[58,68–70] of **Ni-6** was achieved by treatment with an oxidant such as ferrocenium hexafluorophosphate, affording the azetidine product **6a** (Fig. 4c). The reaction of net CO extrusion from $\gamma$-lactam **6** giving azetidine **6a** would be highly endergonic ($\Delta G°$(DFT) = + 32.5 kcal/mol), and the current example highlights the unique potential in contra-thermodynamic decarbonylation reactions through isolated organonickel reagents[58]. Solid **Ni-6** also underwent fast hydrolysis under air affording the ring-opened amino alcohol **6b**, another synthetically relevant motif. In addition, the thermally induced C(sp$^2$)–N bond reductive elimination of **Ni-10** successfully afforded the indoline product **10a** at 90 °C. The versatile reactivity including CO exchange, decarbonylative hydrolysis and ring contraction for the aliphatic $\gamma$-lactam **6** that proceeded through a single intermediate **Ni-6** at RT clearly demonstrates the synthetic potential of the isolable organonickel reagents.

With the insights gathered from the organometallic and DFT studies, the skeletal metalation strategy was applied to structurally complex molecules for proof-of-concept single-atom editing. Azasteroids, such as **19**, are an extensively studied and clinically used class of 5α-reductase inhibitors for the treatment of benign prostatic hyperplasia[71]. The direct editing of the carbon framework could rapidly provide valuable, skeletally distinct derivatives for biological testing. In addition, the selective incorporation of a carbon isotope into a carbonyl group of azasteroids could provide useful new methods for their radiolabelling[65,67] and metabolism studies[72]. The use of stoichiometric metal in highly challenging reactions could also provide significantly improved outcomes, especially in drug discovery research where the successful first-pass reaction significantly outweighs reagent costs[73]. When **19** was subjected to three equivalents of (IPr)Ni($\eta^6$-PhMe) and 50 °C heating, full consumption of **19** was observed after 16 h along with the clean formation of the deinserted nickellacycle **Ni-19** indicated by $^1$H NMR analysis (Fig. 4d). Remarkably, the activation of the lactam C–N bond by (IPr)Ni($\eta^6$-PhMe) was highly selective over competing reactive positions including the ester C–O bond[74] and the $\alpha$-proton of the ester[75]. Oxidatively induced C(sp$^3$)–N reductive elimination from **Ni-19** was achieved by adding I$_2$, affording the ring-contracted azasteroid derivative **19a** with no perturbation of the absolute stereochemistry indicated by X-ray crystallographic and NMR analysis. This discovery showed a powerful method to selectively excise a CO molecule from a lactam ring of stereochemically complex molecules in only two steps. A reported synthesis of the same core structure involved six steps from testosterone benzoate with deliberate ring-opening and closing reactions[76]. Furthermore, the selective carbon isotope exchange of **19** was also achieved by treating **Ni-19** with three equivalents of $^{13}$CO$_{(g)}$ generated from $^{13}$COgen[77], affording the $^{13}$C-labelled azasteroid $^{13}$C-19 with 80($\pm$1)% $^{13}$C-incorporation from mass spectrometry analysis. This result demonstrated a new strategy for late-stage carbon isotope exchange of the carbonyl group in aliphatic lactams, which might prove challenging to perform by established methods[65,67]. Metalation of azasteroid **20**, the $\alpha$, $\beta$-unsaturated variant of **19**, occurred readily at RT affording the corresponding vinyl nickellacycle and treatment with $^{13}$CO successfully afforded the labelled product $^{13}$C-20 (Supplementary Fig. 30). Notably, the ring contraction of **19** could also be performed in a one-pot fashion without the need for intermediate isolation. These results clearly demonstrate the high synthetic promises offered by this new class of organometallic reagents accessed through direct carbonyl-to-metal-exchange logic.

In conclusion, we have introduced the design and synthesis of a versatile class of organonickel reagents from lactams for their direct single-atom editing. Key to this approach is the selective endocyclic C–N bond activation mediated by Ni(0) under mild reaction

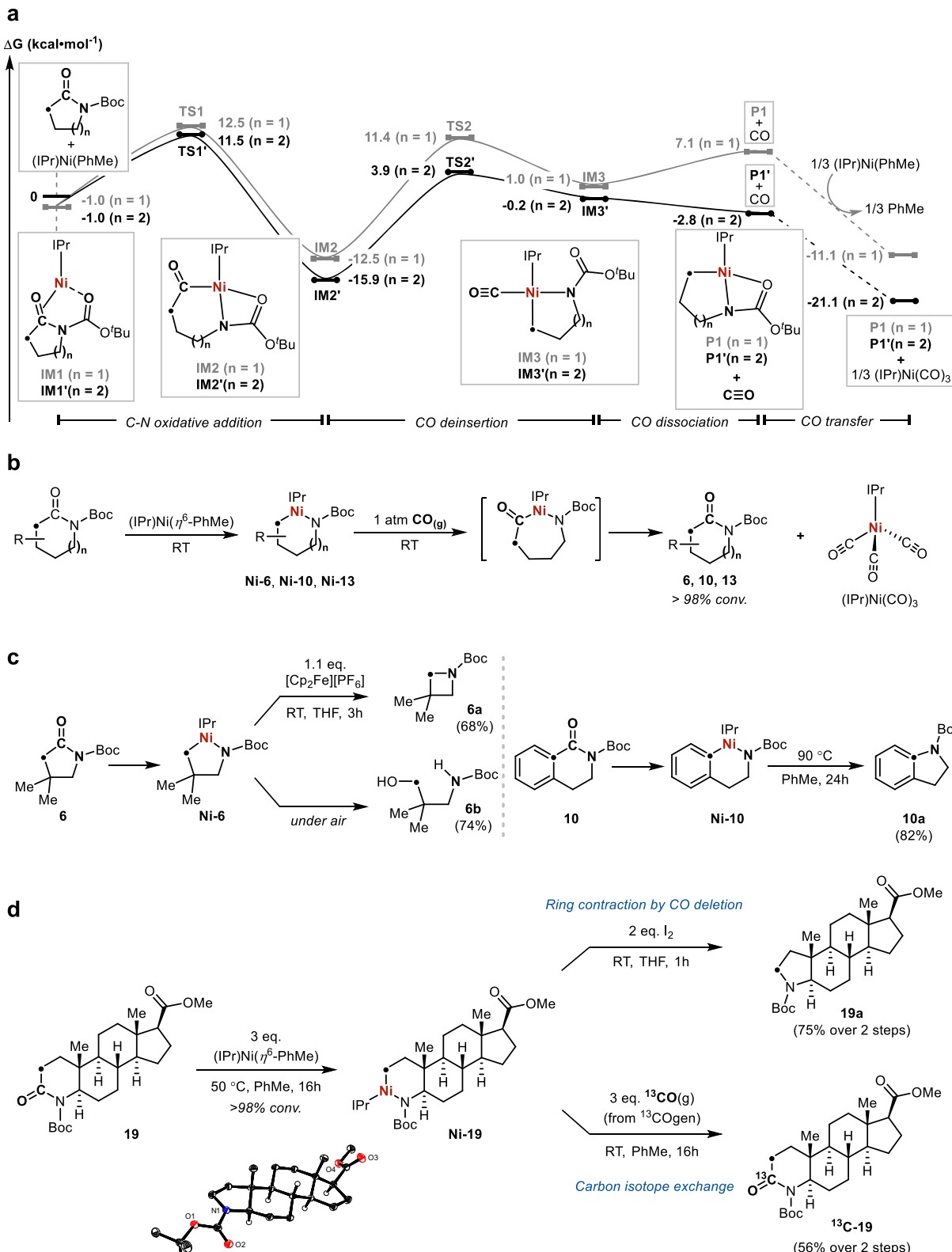

**Fig. 4 | DFT studies and synthetic transformations. a** Computational studies of elementary steps in the skeletal metalation of γ- and δ-lactams **1** and **2** at the PBE0-D3BJ/CPCM(toluene)-def2-TZVP-def2-QZVP(Ni)//BP86-D3BJ/CPCM(toluene)-def2-SVP-def2-TZVP(Ni) level of theory. **b** Reversible C–N oxidative addition and CO deinsertion enable the exchange of lactam carbonyl groups with exogeneous $CO_{(g)}$. **c** Reductive elimination of C($sp^3$)−N and C($sp^2$)−N bonds from nickellacyles and ring-opening hydrolysis. **d** Skeletal metalation of azasteroid **19** and application in decarbonylative ring contraction and carbon isotope exchange with [13]CO. X-ray structure of **19a** shown with 50% thermal ellipsoids and H atoms omitted except for those connected to carbon stereocentres.

conditions. The surgical editing of a lactam carbonyl in azasteroids also demonstrates the application of this strategy to the rapid remodelling of lactam cores in bioactive molecules, underscoring the broad potential of metal-carbon exchange logic in organic synthesis.

## Methods

### General procedure for the carbonyl-to-nickel exchange of lactams and isolation of nickellacycles

In an argon-filled glovebox, a 150 mL Schlenk flask was charged with (IPr)Ni($\eta^6$-PhMe) (0.2 mmol, 1 equiv.), the corresponding lactam (0.2 mmol, 1 equiv.), 5 mL toluene, a stir bar and sealed. The reaction was stirred under room temperature for 16 h. The volatiles were then removed under vacuum on the Schlenk line. The Schlenk flask containing the solid residue was brought back into the glovebox. To the residue was added 1 mL $n$-hexane and the content was filtered. The precipitate was washed with 0.5 mL $n$-hexane twice and dried under vacuum to afford the corresponding nickellacycle.

## Data availability

X-ray crystallographic data for compounds **Ni-1**, **Ni-2**, **Ni-3a/Ni-3b**, **Ni-4**, **Ni-5**, **Ni-7a/Ni-7b**, **Ni-8**, **Ni-9**, **Ni-10**, **Ni-11**, **Ni-12**, **Ni-13**, **Ni-14**, **Ni-15**, **Ni-2-py**, **Ni-2-Ts**, (**IPrNi**)$_2$ and **19a** have been deposited at the Cambridge Crystallographic Data Centre, under deposition numbers CCDC 2227410-2227427. The data can be obtained free of charge via www.ccdc.cam.ac.uk/structures, or by emailing data_request@ccdc.cam.ac.uk, or by contacting The Cambridge Crystallographic Data Centre, 12 Union Road, Cambridge CB2 1EZ, UK; fax: +44 1223 336033. Cartesian coordinates of the DFT-optimised structures are provided in a source file. All additional experimental and computational data are included in the Supplementary Information and are also available from the authors upon request. Source data are provided with this paper.

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

## Acknowledgements

Assistance from the Small Molecule Crystallography Centre (SMoCC), the NMR service and the Molecular and Biomolecular Analysis Service (MoBiAS) at ETH is gratefully acknowledged. We thank the Morandi group for critical proofreading of the manuscript. H.Z. acknowledges the ETH Fellowship for support. D.T.E. acknowledges a fellowship from the Stipendienfonds der Schweizerischen Chemischen Industrie (SSCI). We further acknowledge financial support by the Swiss National Science Foundation (SNSF 184658), the ETH Zürich, and the European Research Council under the European Union's Horizon 2020 research and innovation programme (Shuttle Cat, Project ID: 757608).

## Author contributions

H.Z. conceived the project. B.M. supervised the research. H.Z., D.T.E., V.C.M.G., L.K., M.Z.S. and N.T. conducted the experimental work and analysed the data. D.T.E., P.F. and V.C.M.G. conducted the computational studies. H.Z. and B.M. wrote the manuscript with input from all authors.

## Competing interests

The authors declare no competing interests.
