## [Peer Review File · Nature Communications]

Skeletal Metalation of Lactams through a Carbonyl-to-Nickel-Exchange LogicEditorial Note: This manuscript has been previously reviewed at another journal that is not operating a transparent peer review scheme. This document only contains reviewer comments and rebuttal letters for versions considered at Nature Communications.

Reviewers' Comments:

Reviewer #1:

Remarks to the Author:

I reviewed this article at Nature Synthesis - I am surprised and a little disappointed that this work ended up being transferred down from that journal, but nonetheless I feel that the authors have addressed all of my concerns. This work should be published as is, in my opinion.

Briefly:

I remain very much in support of the core idea here. It is unique and valuable.

The authors have added some discussion of the origins of the structural effects, and while I would have liked to see more, I understand that they may not have a full understanding yet.

The DFT issues have been addressed satisfactorily, including revised barriers, a statement regarding CO dissociation, and the inclusion of the level of theory.

The missing citations have been added.

The title is now grammatically consistent.

Reviewer #2:

Remarks to the Author:

Since I served as a reviewer of this manuscript when it was initially submitted to a sister journal, allow me to repeat the strength of this work:

This paper describes the concept of 'metal-carbon exchange' and provides a proof-of-concept study by demonstrating the synthesis of cyclic organonickel complexes from lactams via decarbonylation. The strong point of this work is the uniqueness of the concept. Although single atom editing is now well-recognized as an emerging concept in synthetic organic chemistry (Nature Synth. 2022, 1, 352), replacing a carbon fragment with a metal is a highly original, unprecedented idea. This new concept was shown to work with several lactam derivatives, which could subsequently be transformed into ring-contracted piperidines and ¹³C-labelled lactams. The reaction mechanism was also investigated theoretically, thereby providing deeper understanding of the effect of the Boc group and ring size. The only concern at this time was the limited scope of the transformations that the nickelacycle can undergo (decarbonylation and the reinsertion of CO). Therefore, this reviewer feels that advantage of isolable nature of the nickel intermediate complexes was not fully demonstrated. By these reasons, publication of this work in Nature Communications is not recommended in its current form.